# Flexible and Highly Sensitive Pressure Sensors Based on Microstructured Carbon Nanowalls Electrodes

**DOI:** 10.3390/nano9040496

**Published:** 2019-04-01

**Authors:** Xi Zhou, Yongna Zhang, Jun Yang, Jialu Li, Shi Luo, Dapeng Wei

**Affiliations:** 1Chongqing Key Laboratory of Multi-Scale Manufacturing Technology, Chongqing Institute of Green and Intelligent Technology, Chinese Academy of Sciences, Chongqing 400714, China; zhouxi16@mails.ucas.edu.cn (X.Z.); zhangyn@cigit.ac.cn (Y.Z.); jialuli_hit@163.com (J.L.); luoshi15@mails.ucas.ac.cn (S.L.); 2University of Chinese Academy of Sciences, Beijing 100049, China; 3Physics Department, Science School, Harbin Institute of Technology, Harbin 150001, China

**Keywords:** carbon nanowalls, high sensitivity, irregular surface morphology, multipixel pressure sensor, sensing imaging

## Abstract

Wearable pressure sensors have attracted widespread attention in recent years because of their great potential in human healthcare applications such as physiological signals monitoring. A desirable pressure sensor should possess the advantages of high sensitivity, a simple manufacturing process, and good stability. Here, we present a highly sensitive, simply fabricated wearable resistive pressure sensor based on three-dimensional microstructured carbon nanowalls (CNWs) embedded in a polydimethylsiloxane (PDMS) substrate. The method of using unpolished silicon wafers as templates provides an easy approach to fabricate the irregular microstructure of CNWs/PDMS electrodes, which plays a significant role in increasing the sensitivity and stability of resistive pressure sensors. The sensitivity of the CNWs/PDMS pressure sensor with irregular microstructures is as high as 6.64 kPa^−1^ in the low-pressure regime, and remains fairly high (0.15 kPa^−1^) in the high-pressure regime (~10 kPa). Both the relatively short response time of ~30 ms and good reproducibility over 1000 cycles of pressure loading and unloading tests illustrate the high performance of the proposed device. Our pressure sensor exhibits a superior minimal limit of detection of 0.6 Pa, which shows promising potential in detecting human physiological signals such as heart rate. Moreover, it can be turned into an 8 × 8 pixels array to map spatial pressure distribution and realize array sensing imaging.

## 1. Introduction

Wearable electronics [1,2,3,4,5] and artificial electronic skins [6,7,8] have been developing rapidly. Flexible pressure sensors [9,10], as the core functional components of the aforementioned devices, have drawn much attention to human–machine interfaces [11,12,13], motion detection [14,15], and physiological signal monitoring [16]. Pressure sensors provide abundant physical signals, such as blood pressure [17] and heart rhythm [18]. Different application scenarios require different pressure ranges that the pressure sensors should specialize in, for example, acoustic pressure is lower than 10 Pa [19]; daily activities like gentle touching, grasping, and health monitoring are less than 10 kPa; while weight measurement and manipulation of robot arm require pressure in the 10–100 kPa range [20]. Although significant achievements have been accomplished to fabricate pressure sensors with high sensitivity, crucial issues remain regarding market promotion, such as high compatibility, comfortable attachment, and micro integration.

The microstructure design of electrodes is an indispensable process for flexible pressure sensors. Many works utilizing microstructures or nanostructures to regulate sensitivity have been reported. For example, Bao [21] and co-authors fabricated a pressure sensor with a regular micro-pyramid and micro-hair structure whose sensitivity was quite low (0.58 kPa^−1^). Ren [22] and his co-authors have reported that the sensitivity with randomly distributed spinosum structures is better than the regular structure such as pyramid [23], hemisphere, and nanowire, while the relatively long response time of 120 ms limited its applications in fast responding cases. Natural irregular structures are mainly focused on the advantage that designers are free from fabricating complicated artificial structures. Ren [22] and his co-authors transferred the structure of abrasive paper to polydimethylsiloxane and fabricated a piezoresistive pressure sensor. Guo [24] and co-authors reversely moulded the micro column structure on a calathea leaf as the irregular microstructure. Unfortunately, there are some disadvantages using the natural templates for the plant leaves are limited by seasonal factors. Moreover, these methods have difficulties in implementing large area array sensing. The development of fabricating materials with various morphologies and microstructures contributes greatly to the advanced pressure sensors. Particularly, carbon-based materials such as carbon tubes [3,25], carbon black [26], and graphene [11,18,27,28,29] have played an important role in both sensing elements and conductive fillers. Carbon black combined with other materials has been used as the main material to fabricate sensitive electrodes, while the conductivity may degrade along the life cycle. Carbon nanowalls (CNWs), a type of three-dimensional (3D) carbon nanomaterial, have shown remarkable electrical and mechanical properties due to their hexagonally packed sp2-hybridized carbon atoms and 3D stack structure. CNWs can be hybrid with flexible organic polymers such as polydimethylsiloxane (PDMS) and show excellent electrical stability. In addition, CNWs can be grown in a patterned array on a silicon wafer using a mask. Therefore, these merits make CNWs a promising material in the field of flexible pressure sensors.

Similar to the epidermis of human skin, the surface of unpolished silicon has random microstructures. In this work, we present a wearable resistive pressure sensor with a simple and stable structure that is based on the CNWs electrodes fabricated through rough silicon wafer mold. CNWs are conformally grown on an unpolished silicon wafer in plasma enhanced chemical vapor deposition (PECVD) system. Then the microstructured CNWs on the rough silicon surface are transferred onto the flexible PDMS substrate. The CNWs/PDMS composite electrodes show sufficient mechanical strength and stable electrical conductivity. The fabrication process is completed by a face-to-face assembly of two CNWs/PDMS composite conductors. Our pressure sensor obtains high sensitivity, stable data output at various pressures, and fast response speed. Besides, our flexible sensor shows mechanical durability (1000 cycles) with reversible on and off load test. Due to the prominent performance of CNWs/PDMS electrodes with an irregular microstructure, the pressure sensor shows great potential in micro-weight detection and human physiological signal detecting such as a difference in heart rate between normal condition and that after exercise. It is also notable that the multi-pixel resistive pressure sensor based on CNWs/PDMS composite array presents excellent pressure imaging properties.

## 2. Materials and Methods

Fabrication of CNWs/PDMS Pressure Sensors. Commercial silicon wafer with an unpolished side was utilized as template. For a single device, carbon nanowalls were grown in a tube furnace with plasma-enhanced chemical vapor deposition (PECVD) system. The growth temperature was 750 °C; the flow rates of methane and hydrogen were 6 sccm and 4 sccm, respectively; and the radio frequency (RF) power was 200 W. The polydimethylsiloxane (Sylgard 184, Dow Corning, Midland, MI, USA) prepolymer was prepared by mixing the base silicone gel well with a curing agent at a weight ratio of 10:1. After being subjected to vacuum for 30 min in a chamber, the mixture was poured onto the rough side with carbon nano-walls to naturally form a thin film. Then, it was cured on the hot plate at a temperature of 80 °C for 2 h. The CNWs/PDMS sample was next cut into rectangular shape and connected to a copper wire at one end of the piece with silver paste. Face-to-face packaging of the CNWs/PDMS samples completed the fabrication process.

Characterization of Morphology and Performance of CNWs/PDMS Pressure Sensors. The morphologies of the fabricated sensors were characterized by atomic force microscopy (Dimension EDGE, Bruker, Karlsruhe, Germany) and field emission scanning electron microscope (SEM) (JSM-7800F, JEOL, Tokyo, Japan). Raman spectra were carried out using a laser of 532 nm (InVia Reflex, Renishow, Gloucestershire, United Kingdom). The loading of applied force was carried out with a testing machine (ETM503A, WANCE, Shenzhen, China), and the electrical signals of the pressure sensors were recorded at the same time by a digital multimeter (2450, Keithley, Beaverton, OR, USA).

## 3. Results and Discussion

Microstructures always improve the performance of pressure sensors. Consequently, a simple way of fabricating microstructure was introduced, which is illustrated in Figure 1a, where rough silicon wafer with irregular morphology provided a three-dimensional surface for CNWs growing. Because the adhesion between CNWs and PDMS is stronger than that between CNWs and silicon, CNWs could be easily transferred onto PDMS and the composite could act as a flexible electrode. The SEM images of the microstructure of rough silicon wafer, conformally grown CNWs on the wafer and the CNWs transferred onto the PDMS, are clearly shown in Figure 1b. Appendix A show the 3D stacked carbon structures that formed from conformally grown CNWs on the irregular surface. After transferring the CNWs to the PDMS, the shape of the template was retained. From Appendix A, we can clearly see that CNWs were firmly attached to the surface of PDMS. Highly sensitive and flexible pressure sensors were obtained by a simple face-to-face package process; the optical image of flexible pressure sensor is shown in Figure 1c. The surface topography of silicon mold is shown in Appendix A, and the SEM image of the silicon mold is shown in Appendix A. For more details of the rough silicon wafer with CNWs, atomic force microscope (AFM) was used to measure the surface topography, as shown in Figure 1d; the area sampled by the probe of AFM was 50 μm × 50 μm, and the difference between the maximum and minimum spinosum surface heights was about 6 μm. The surface topography of the electrode is shown in Appendix A, which indicates the shape of the silicon mold is well preserved.

After assembly, several conducting channels between protrusions on each electrode with irregular surface morphology emerge, as shown in Figure 1e. The working principle of the high-sensitivity pressure sensor is changing the contact resistance between the upper and lower electrodes. When the sensor is in its initial state (P1 = 0), only a small amount of conduction channel exists between the two electrodes, and the contact resistance is high. When pressure is applied to the sensor (P3 > P2 > P1), the pressure reduces the space between the upper and lower electrodes, and more protrusions on the electrodes contact each other, forming more conductive channels; hence, the conductivity between the upper and lower electrodes improves. Owing to the three-dimensional structures of CNWs, the extrusion deformation of the PDMS substrate does not destroy the carbon nano-structure, and the electrodes can maintain a stable electrical conductivity. The bonding form of carbon was characterized by Raman spectrometer, as shown in Appendix A. G peak is the main characteristic peak of graphene, and G’ peak illustrates the three-dimensional stacking mode of CNWs. Figure 1f shows the relation between pressure and conductivity of the sensor, which is that the electrical conductivity of the device grows with increasing pressure.

Figure 2a illustrates the characterization system. Precise pressure is applied by the force controller. The pressure probe feeds back the pressure values to the system in real time. Current signals are collected by the analyzer. The pressure sensor made from CNWs-coated rough silicon wafer exhibits high sensitivity and large linearity. Figure 2b shows the relative current variation when the applied pressure is gradually increased up to 10 kPa. The current change in low pressure range (0–1 kPa) is shown in the inset in Figure 2b. There are two distinguishable sections to illustrate different pressure sensitivity. The sensitivity, which is expressed as S = (I − I_0_)/(I_0_P), is usually used to characterize the performance of the pressure sensor, where I_0_ and I are the initial current and current under applied pressure, respectively. According to the expression, the calculated sensitivity of the CNWs/PDMS pressure sensor is 6.64 kPa^−1^ in the range of 0–200 Pa and 1.26 kPa^−1^ in the range from 200 Pa to 1000 Pa. In the pressure range of 1–10 kPa, the sensitivity of the sensor is 0.15 kPa^−1^. Because of the increasing contact area, the current between two electrodes experienced an overall rise through the increment of pressure. Eventually, the pressure response of the sensor was almost saturated at 10 kPa.

The highly sensitive CNWs/PDMS pressure sensor shows low hysteresis property. Being applied pressure continuously increasing from 0 to10 kPa, the conductivity of the device gradually improves. When the pressure is reduced from 10 kPa to 0 kPa, the electrical conductivity of the device returns to the initial state. The forward and backward curves basically coincide, as shown in Figure 2c, which indicates that our pressure sensor fully can be restored after deformation. In order to further measure the hysteresis of the device, various applied pressures ranging from 0 Pa to 200 Pa were gradually applied to the sensor, and then the pressures were unloaded in turn. As can be seen from the chart in Figure 2c, the conductivity of the sensor quickly restores itself to the initial state after deformation. The fast response performance is shown in Figure 2d, which presents a response time of 30 ms upon loading and unloading the input pressure of 50 Pa. This sensor exhibits a fast response speed, because the CNWs on the upper and lower electrodes surfaces do not stick together after contact. When the external pressure is removed, the contact section can be separated within a short period of time due to the recovery of deformation. The hysteresis of the response time is mainly attributed to elastic recovery of the PDMS substrate.

Bending and stretching deformation frequently occur when the tactile sensor is attached to the human skin. Therefore, the stability of the tactile sensor after bending and tensile deformation is significantly important. We tested the stability of the CNWs electrode after 1000 times bending deformation. A slight crack appeared on the surface of the electrode as shown in Appendix A. The bending angles increased from 0° to 45°, as shown in Appendix A. The relationship between the bending angle and the resistance value is shown in Appendix A. After 1000 bends, the CNWs electrode still maintains excellent stability, and the resistance value of the electrode can return to the initial state when the deformation is restored, as shown in Appendix A. It is worth mentioning that the thickness of our CNWs/PDMS electrode is only 465 µm, as shown in Appendix A. In order to test the stability of the CNWs electrode after undergoing tensile deformation, we performed tensile experiments with 10%, 30%, and 50% deformation of the electrodes. As shown in Appendix A, the resistance value of the electrode increases with the deformation, due to the induced cracks under tensile strain. However, although the surface of the electrode cracked after 1000 stretches, as shown in Appendix A, the conductivity of the electrode returned to the initial state after the deformation was restored. The tensile deformation causes temporary damage to the CNWs network structure, and the conductive network can be reformed after the releasing of deformation, as shown in Appendix A.

To investigate the repeatability of the pressure sensor under different pressure conditions, several pressure values from 10 Pa to 200 Pa were repeatedly applied to the sensor to monitor the corresponding variation in electrical resistance of the pressure sensor in real time, as shown in Figure 3a. Furthermore, the high-sensitivity pressure sensor has excellent frequency response. As shown in Figure 3b, the frequency was increased from 0.5 Hz to 2 Hz. Owing to the extremely short response time of the device, the sensor rapidly responded to the pressure of different frequencies in real time. Besides, the relative resistance variation of the CNWs/PDMS pressure sensor exhibits high stability and durability under periodic pressure loading, as shown in Figure 3c. Magnified sensor responses extracted from Figure 3c are shown in Figure 3d, which clearly shows that there is no obvious degeneration during the whole cyclic process.

Owing to high sensitivity, the CNWs/PDMS pressure sensor exhibits tremendous utility for the detection of human physiological signals and motion activities. Firstly, we demonstrated the sensitive detection performance of the sensor by dropping tiny droplets onto it. As shown in Figure 4a, the volume of water droplets was 6 μL, and the pressure generated on the sensor surface was around 0.6 Pa. We can see from Figure 4b that the high-sensitivity pressure sensor responded to tiny droplets that were continuously dripping. The value of the pressure ranges from 0.6 to 2.4 Pa. As can be seen from Figure 4b, the curve of the relative current value is stepped because the sensor has good linearity in this pressure range, and the slight hysteresis allows the CNWs/PDMS sensor to respond quickly to small pressures. Based on the high sensitivity and fast response speed, our sensors can be used to detect human physiological signals. Heart rate is the frequency of the cardiac cycle caused by myocardial contraction and pumping blood from the chamber into the artery, which is one of the important parameters to assess a person’s physical and mental state. Figure 4c shows a heart rate signal gathered by CNWs/PDMS pressure sensor, which was attached on the wrist by medical tape to detect the pulse. It can be seen from Figure 4c that the heart rate is clearly measured by the pressure sensor from the radial artery of the wrist. The heart rate can also reflect the motion state of the person. It is worth noting that the muscle cells of the human body need a lot of oxygen and energy, and the frequency and intensity of the heart beat will increase to ensure the stability of the body after exercise. The curve in Figure 4c clearly shows the difference in heart rate between normal heart rate and that after exercise. A magnified comparison of individual peaks is shown in Figure 4d. The main peak (P1) of heart rate after exercise is significantly greater than that before the exercise, and the tidal wave peak (P2) of heart rate after exercise is also significantly higher than the point before exercise, which indicates an increase in the heart beat intensity brought on by exercise. We can also clearly see from Figure 4d that the heart beats significantly faster after exercise than it does in a static state. As to one heart rate cycle, the one after exercise is nearly 0.1 s shorter than that before exercise. It can be concluded that the CNWs/PDMS pressure sensor shows excellent capability for healthcare monitoring of vital signals.

A single sensor can only achieve single point detection and cannot meet the needs of large area detection. To realize mapping the pressure like real human skin, a two-dimensional sensors array is manufactured. The fabrication process of the flexible CNWs/PDMS sensors array is schematically illustrated in Appendix A. In order to achieve the patterned growth of CNWs, we used a metal plate with array holes (8 × 8, pixel size of 5 mm × 5 mm) as a growth mask so that the carbon atoms were only deposited in the pores of it. The optical image of the CNWs array on silicon substrate is shown in Figure 5a. Then, we spin-coated PDMS onto silicon substrate with CNWs to peel off the CNWs and form a flexible sensor array. Then, we packaged the flexible sensor array in a flexible printed circuit (FPC), as shown in Figure 5b. The acquisition circuit was responsible for converting the analog signal of the change in resistance caused by the pressure into a digital signal. The STM32F103VCT6 chip of STMICROELECTRONICS was used as the core controller for the acquisition circuit. Figure 5c illustrates the system schematic. The details of the signal processing circuit are shown in Figure 5d. Array-type tactile sensors still responds well to a single point, as shown in Figure 5e. One can see that there is almost no crosstalk between adjacent pixels. The mapping of output resistance signals can reflect the metal letter in the shape of “E” placed on the sensors array well, as shown in Figure 5f. In addition, we verified the robustness of the sensors array using different letters (CIGIT), and the result is shown in Appendix A. These results indicate that the CNWs/PDMS pressure sensors can be made to arrays and perform like our skin to map the shapes of external pressures, enabling them to be potential devices in the wearable electronics and intelligent robot field. Importantly, the fabricated flexible sensor array with small pixel size can be applied to different scenes by changing the size and shape of the array. Thus, our reported pressure sensor based on microstructured CNWs is compatible for further miniaturization and scalability.

## 4. Conclusions

In summary, owing to the conformally growing technique and the randomly distributed microstructures on the unpolished silicon surface, we introduce an easy approach to fabricate high-performance flexible pressure sensors with irregularly patterned microstructured electrodes. The CNWs microstructures endow the pressure sensor with ultrahigh sensitivity, which is 6.64 kPa^−1^ in the pressure range <0.2 kPa, 1.26 kPa^−1^ in the range 0.2 kPa to 1 kPa, and 0.15 kPa^−1^ at 10 kPa. The sensor also presents an ultralow limit of detection of 0.6 Pa due to the surface modification of PDMS by carbon nanowalls. Furthermore, the response time is as short as 30 ms. These indicators show the pressure sensor has an excellent comprehensive performance compared with other pressure sensors with patterned surfaces. The pressure sensor exhibits immense potential in the detection of physiological activities and can be used as human healthcare monitoring. Our tactile sensor arrays can be designed in a variety of shapes to suit different application needs. In future smart health devices, our tactile sensor arrays can be placed in the insole for gait detection and gait pressure distribution image. The characteristics of high-sensitivity, flexibility, and good repeatability provide the possibility to fit on smart wearable devices in the future applications.

## Figures and Tables

**Figure 1 nanomaterials-09-00496-f001:**
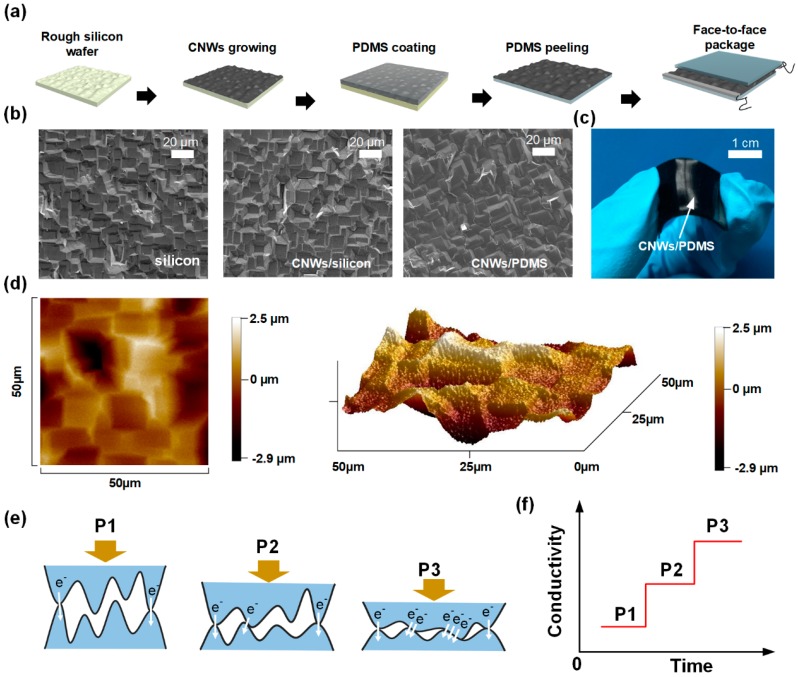
Resistive pressure sensor based on flexible carbon nanowalls (CNWs)/ polydimethylsiloxane (PDMS) composite electrodes with irregular surface morphology. (**a**) Schematic illustration of the fabrication process of the flexible pressure sensor; (**b**) scanning electron microscopy (SEM) image of the rough silicon surface, and CNWs on rough silicon surface and the CNWs/PDMS surface; (**c**) physical image of the fabricated sensor; (**d**) the surface topography of the silicon wafer with CNWs, which is detected by AFM. (**e**) Schematic illustration of the basic working principle of the sensor; (**f**) is the diagram of the electrical conductivity of the device over time.

**Figure 2 nanomaterials-09-00496-f002:**
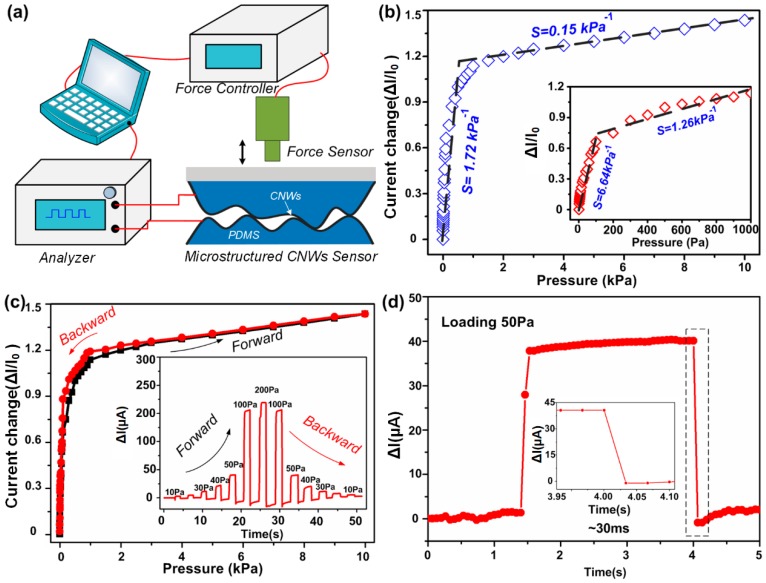
Pressure sensing performance of a flexible pressure sensor based on irregularly roughened CNWs/PDMS composite electrodes. (**a**) Mechanical property test system. (**b**) Change in the electrical resistance of the sensors as a function of applied pressure (inset: magnified sensor responses representing pressure-sensitivities). (**c**) Illustration of the relationship between force and current during loading and unloading pressure; the two curves are basically coincident, indicating that the device has a small hysteresis (inset: current versus time during application of pressure and unloading pressure). (**d**) Magnified sensor responses extracted from (**c**) to show response time.

**Figure 3 nanomaterials-09-00496-f003:**
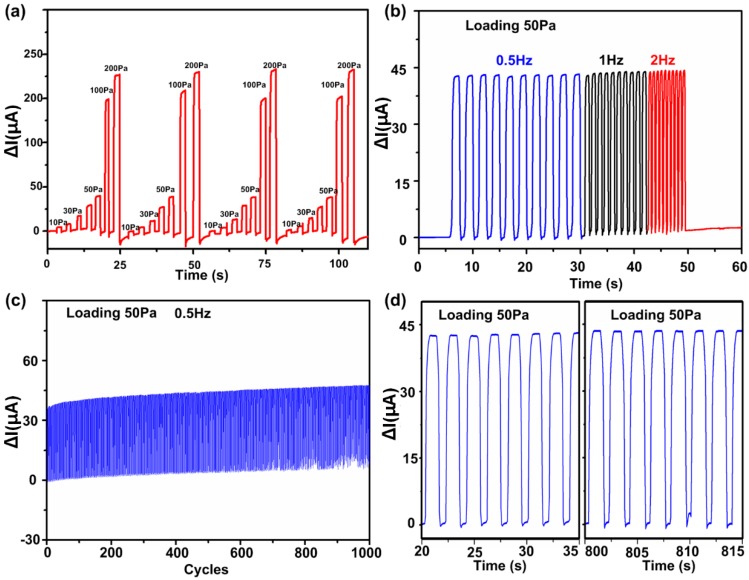
Stability of the high sensitivity pressure sensor under different types of pressure. (**a**) Sensor responses under repetitive pressure loading and unloading cycles with different values from 0 Pa to 200 Pa. (**b**) Current curve of the sensor under the pressure of different frequencies from 0.5 Hz to 2 Hz. (**c**) Variation in the electrical resistance of the sensor under 1000 loading and unloading cycles with a maximum pressure of 50 Pa, and (**d**) magnified waveforms extracted from (**c**) at several time intervals, representing sensor stable response for repeated operations.

**Figure 4 nanomaterials-09-00496-f004:**
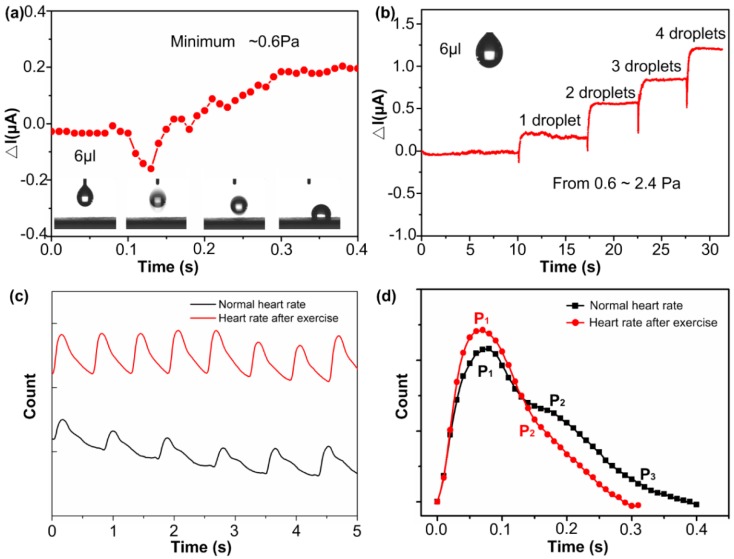
High sensitivity pressure sensor applicated in minor pressure detection. (**a**) Illustration of the change in the current of a water droplet as it falls onto a sensor; the capacity of a drop of water is 6 μL, and the pressure generated is approximately 0.6 Pa—this is also the detection limit of the sensor. (**b**) The current change while continuously dripping water onto the sensor. (**c**) The normal heat rate compared with the heat rate after exercise in current change; (**d**) is the enlarged view of one of the waveforms.

**Figure 5 nanomaterials-09-00496-f005:**
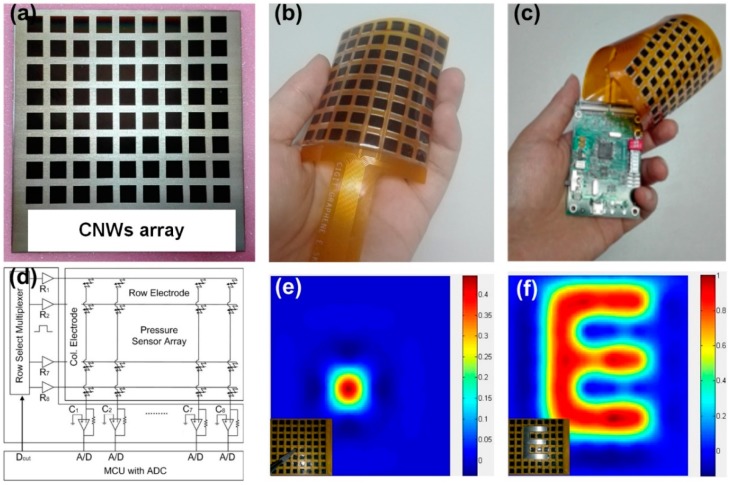
Spatial mapping of pressure. (**a**) Physical image of the patterned growth of carbon on a silicon substrate. (**b**) Sensor array packaged with flexible printed circuit (FPC). (**c**) Sensor array imaging system with analysis circuit. (**d**) Current signal extraction and processing circuit. (**e**) Response of an array of sensors to a single point of pressure. (**f**) A metal stamp in the shape of “E” placed on the sensors array and corresponding mapping of the current changes.

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
