# Peer review of "Flexible and Highly Sensitive Pressure Sensors Based on Microstructured Carbon Nanowalls Electrodes"

_nanomaterials, 2019, doi:10.3390/nano9040496_

Round 1

Reviewer 1 Report

This study reports a sensitive, simply fabricated wearable resistive pressure 16 sensor based on three-dimensional microstructured carbon nanowalls (CNWs) embedded in a  polydimethylsiloxane (PDMS) substrate. The reported pressure sensor exhibits minimal limit of detection of 0.6 Pa, which shows promising potential in detecting human physiological signals such as heart rate.

-       I suggest authors to include below recent review papers in their reference list, in addition to Ref#1-3:

Dagdeviren, C., Joe, P., Tuzman, O. L., Park, K., Lee, K. J., Huang, Y., Rogers, J. A., “Recent Progress in Flexible and Stretchable Piezoelectric Devices for Mechanical Energy Harvesting, Sensing and Actuation”, Extreme Mechanics Letter, 9(1), 269-281, 2016.

Dagdeviren, C., Li, Z., Wang, Z. L., “Energy Harvesting from the Animal/Human Body for Self-Powered Electronics”, Annual Review of Biomedical Engineering, 19, 85-108, 2017.

-       I suggest authors to include below paper in their reference list, in addition to Ref#6-7:

Persano, L., Dagdeviren, C., Su, Y., Zhang, Y., Girardo, S., Pisignano, D., Huang, Y., Rogers, J. A., “High Performance, Flexible Piezoelectric Devices Based on Aligned Arrays of Nanofibers of Poly[(vinylidenefluoride-co-trifluoroethylene]”, Nature Communications, 4, 1633, 2013.

-       I suggest authors to include below paper in their reference list, in addition to Ref#12:

-        

Dagdeviren, C., Su, Y., Joe, P., Yona, R., Liu, Y., Kim, Y. S., Damadoran, A. R., Huang, Y. A., Xia, J., Martin, L. W., Huang, Y., Rogers, J. A., “Conformable Amplified Lead Zirconate Titanate Sensors with Enhanced Piezoelectric Response for Cutaneous Pressure Monitoring”, Nature Communications, 5, 4496, 2014.

-       The use of more junction words (such as thus, hence, therefore, since, whereas, etc.) and sentences can help the reading be more smooth and the paragraphs to more easily merge into one another.

-       There is a lack of device design studies regarding miniaturization or scalability, which is relevant to the current challenges in devices for wearable and implantable or non-linear surfaces or structures applications.

-       How durable is the device (Figure 3) mechanically? A repetitive comprehensive mechanical pressure test should be conducted to comment on any possible mechanical fatigue. As a support, SEM images of the active layer (i.e., CNWs/PDMS interfaces) surface before and after bending can be given to prove that there has been no cracks or distortion formed in between layers or in overall device structure.

- How thin can the sensor be? Is there any relation between thickness and sensitivity of the sensor?

- For practical usage, how humidity will incorporate into your experimental results? Does humidity have any effect on the mechanical and electrical stability of your sensor especially for trials in Fig. 4?

The conclusion can be made more interesting to the readers due to the impact of this research by giving examples of immediate and future applications that are possible with this technology. Also, it will be informative if the authors could include their future research plan (such as closed-loop feedback with wearable healthcare monitoring).

Author Response

Point 1: I suggest authors to include below recent review papers in their reference list, in addition to Ref#1-3:

Dagdeviren, C., Joe, P., Tuzman, O. L., Park, K., Lee, K. J., Huang, Y., Rogers, J. A., “Recent Progress in Flexible and Stretchable Piezoelectric Devices for Mechanical Energy Harvesting, Sensing and Actuation”, Extreme Mechanics Letter, 9(1), 269-281, 2016.

Dagdeviren, C., Li, Z., Wang, Z. L., “Energy Harvesting from the Animal/Human Body for Self-Powered Electronics”, Annual Review of Biomedical Engineering, 19, 85-108, 2017.

Response 1: Thanks for your kind advice, we have added the above references in the INTRODUCTION section to improve our manuscript.

4.         Dagdeviren, C.; Joe, P.; Tuzman, O.L.; Park, K.-I.; Lee, K.J.; Shi, Y.; Huang, Y.; Rogers, J.A. Recent progress in flexible and stretchable piezoelectric devices for mechanical energy harvesting, sensing and actuation. Extreme Mechanics Letters 2016, 9, 269-281, doi:10.1016/j.eml.2016.05.015.

5.         Dagdeviren, C.; Li, Z.; Wang, Z.L. Energy Harvesting from the Animal/Human Body for Self-Powered Electronics. Annual Review of Biomedical Engineering 2017, 19, 85-108, doi:10.1146/annurev-bioeng-071516-044517.

Point 2: I suggest authors to include below paper in their reference list, in addition to Ref#6-7:

Persano, L., Dagdeviren, C., Su, Y., Zhang, Y., Girardo, S., Pisignano, D., Huang, Y., Rogers, J. A., “High Performance, Flexible Piezoelectric Devices Based on Aligned Arrays of Nanofibers of Poly[(vinylidenefluoride-co-trifluoroethylene]”, Nature Communications, 4, 1633, 2013.

Response 2: Thanks for your kind suggestion, we have added the above reference in the INTRODUCTION section to improve our manuscript.

8.        Persano, L.; Dagdeviren, C.; Su, Y.; Zhang, Y.; Girardo, S.; Pisignano, D.; Huang, Y.; Rogers, J.A. High performance piezoelectric devices based on aligned arrays of nanofibers of poly(vinylidenefluoride-co-trifluoroethylene). Nat Commun 2013, 4, 1633, doi:10.1038/ncomms2639.

Point 3: I suggest authors to include below paper in their reference list, in addition to Ref#12:

Dagdeviren, C., Su, Y., Joe, P., Yona, R., Liu, Y., Kim, Y. S., Damadoran, A. R., Huang, Y. A., Xia, J., Martin, L. W., Huang, Y., Rogers, J. A., “Conformable Amplified Lead Zirconate Titanate Sensors with Enhanced Piezoelectric Response for Cutaneous Pressure Monitoring”, Nature Communications, 5, 4496, 2014.

Response 3: According to your kind advice, we have added the above reference in the INTRODUCTION section to improve our manuscript.

13.       Dagdeviren, C.; Su, Y.; Joe, P.; Yona, R.; Liu, Y.; Kim, Y.S.; Huang, Y.; Damadoran, A.R.; Xia, J.; Martin, L.W., et al. Conformable amplified lead zirconate titanate sensors with enhanced piezoelectric response for cutaneous pressure monitoring. Nat Commun 2014, 5, 4496, doi:10.1038/ncomms5496.

Point 4: The use of more junction words (such as thus, hence, therefore, since, whereas, etc.) and sentences can help the reading be more smooth and the paragraphs to more easily merge into one another.

Response 4: Thanks for your kind suggestion. In revised manuscript, we have added more junction words in the following sentences and made the paragraphs easier to read:

In addition, CNWs can be grown in a patterned array on a silicon wafer using a mask. Therefore, these merits make CNWs a promising material in the field of flexible pressure sensors. (line 65)

Consequently, a simple way to introduce microstructures was illustrated in Figure 1(a), where rough silicon wafer with irregular morphology provided a three-dimensional surface for CNWs growing. (line 101)

Point 5: There is a lack of device design studies regarding miniaturization or scalability, which is relevant to the current challenges in devices for wearable and implantable or non-linear surfaces or structures applications.

Response 5:  Thanks for your kind comments. Indeed, it is very important that both the miniaturization and scalability for wearable and implantable or non-linear surfaces or structures applications, which should be taken into consideration during device designing. Our tactile sensors are characterized by good flexibility, stretchability and lightweight. Importantly, as shown in Figure 5 (a-c), a sensors array with pixel size of 5 mm×5 mm was fabricated for pressure distribution image, which can be applied to different scenes by changing the size and shape of the array.  Thus, our reported pressure sensor based on microstructured CNWs is compatible for further miniaturization and scalability. In revised manuscript, we have added the following sentences in last paragraph of RESULTS AND DISCUSSION section (page 9, marked in red color):

“Importantly, the fabricated flexible sensor array with small pixel size can be applied to different scenes by changing the size and shape of the array. Thus, our reported pressure sensor based on microstructured CNWs is compatible for further miniaturization and scalability.”

Point 6: How durable is the device (Figure 3) mechanically? A repetitive comprehensive mechanical pressure test should be conducted to comment on any possible mechanical fatigue. As a support, SEM images of the active layer (i.e., CNWs/PDMS interfaces) surface before and after bending can be given to prove that there has been no cracks or distortion formed in between layers or in overall device structure.

Response 6: According to your kind guidance, we have supplemented experiments on durability of electrodes. In tactile sensors of wearable devices, it is important to maintain excellent stability after bending and tensile deformation. Our tactile sensor electrodes have undergone 1000 bending experiments with a maximum bending angle of 45°. We obtained the SEM image of the electrode surface after the bending test and the tensile test, respectively. In the experiment, we tested the resistance change of the electrode. As the bending angle increases, the resistance value of the electrode also increases. When the bending angle returns to the initial state, the resistance of the electrode can also return to the original state. In the stretching experiment, our tactile sensor electrode underwent 1,000 cycles of 50% deformation, the deformation of the electrode will bring changes in resistance and surface topography. Similar to the bending deformation, the electrical conductivity of the electrode can also be consistent with the initial state when the deformation is restored. The details of the bending and tensile deformation experiments are presented in new Figure S4 and Figure S5. Therefore, we have added the following contents in the RESULTS AND DISCUSSION section (page 5, marked in red color):

     Bending and stretching deformation frequently occur when the tactile sensor is attached to the human skin. Therefore, the stability of the tactile sensor after bending and tensile deformation is significantly important. We tested the stability of the CNWs electrode after 1000 times bending deformation. A slight crack appeared on the surface of the electrode as shown in (a), (b) and (c) in Figure S4. The bending angles increased from 0 ° to 45 °, as shown in (d) and (f) in Figure S4. The relationship between the bending angle and the resistance value is shown in Figure S4(e). After 1000 bends, the CNWs electrode still maintains excellent stability, and the resistance value of the electrode can return to the initial state when the deformation is restored, as shown in Figure S4(g). It's worth mentioning that the thickness of our CNWs/PDMS electrode is only 465 µm, as shown in Figure S4(f). In order to test the stability of the CNWs electrode after undergoing tensile deformation, we performed tensile experiments with 10%, 30% and 50% deformation of the electrodes. As shown in Figure S5(a)-(c), the resistance value of the electrode increases with the deformation, due to the induced cracks under tensile strain. However, although the surface of the electrode cracked after 1000 stretches, as shown in Figure S5(e)-(f), the conductivity of the electrode returned to the initial state after the deformation was restored. The tensile deformation causes temporary damage to the CNWs network structure, and the conductive network can be reformed after the releasing of deformation, as shown in Figure S5(d).”

Figure S4. (a) Surface morphology of CNWs electrodes in the initial state; (b) Surface morphology after 100 cycles bending; (c) Surface morphology after 1000 cycles bending; (d) and (f) are physical pictures of electrodes at different bending angles; (e) Resistance changes at different bending angles; (g) Resistance changes under bending angle ~ 45 °; (h) Thickness testing of the CNWs electrode.

Figure S5. (a) The physical picture of the electrode tensile test in the initial state; (b) Resistance changes of the electrode under different tensile amounts; (c) Photograph of the CNWs electrode at 50% deformation; (d) Resistance changes of the electrode under 50% deformation; (e) Morphology of the electrode surface before the tensile test; (f) Surface morphology of the electrode after 1000 tensile tests.

Point 7: How thin can the sensor be? Is there any relation between thickness and sensitivity of the sensor?

Response 7: Thank you for your reminding. As shown in Figure S4(h), the thickness of a single CNWs/PDMS electrode is 465 µm. Our pressure sensor is fabricated by two identical electrodes, so the thickness of our pressure sensor is within 1 mm. In our tactile sensors, the key to sensitivity is the random microstructure of the electrode surface, which is less than 10 µm in size. When the device is subjected to external forces, deformation mainly happened on the microstructure of the electrode surface, so we believe that the thickness of the device has little effect on the sensitivity. Above all, we have added the following content marked in red at line 177:

“It's worthy to mention that the thickness of our CNWs/PDMS electrode is only 465 µm, as shown in Figure S4(h).”

Figure S4. (h) Thickness testing of the CNWs electrode.

Point 8: For practical usage, how humidity will incorporate into your experimental results? Does humidity have any effect on the mechanical and electrical stability of your sensor especially for trials in Fig. 4?

Response 8: Thanks for your kind attention. For resistive pressure sensors, humidity has a certain effect on the stability of the sensor. To eliminate this effect, in the sensor array, the electrodes were packaged by thin PET film and sealant. Hence, our device was isolated from water and air, eliminating the effect of humidity.

Point 9: The conclusion can be made more interesting to the readers due to the impact of this research by giving examples of immediate and future applications that are possible with this technology. Also, it will be informative if the authors could include their future research plan (such as closed-loop feedback with wearable healthcare monitoring).

Response 9: Inspired by your kind suggestion, we have improved the conclusions. The most suitable application scenario for flexible tactile sensors is wearable smart electronic devices. In the field of healthcare, electronic skin as a flexible ultra-thin device can be in good contact with human skin and monitor people's health parameters in real time. We can image the pressure distribution with the pixels in the array sensor. We can fabricate the sensor array into various shapes such as insole, which can detect the pressure distribution of the sole of the foot and help the gait correction. With closed loop feedback, the user can make real-time adjustments to the gait. We updated the CONCLUSIONS section as shown below (line 276, marked in red color):

In summary, owing to the conformally growing technique and the randomly distributed microstructures on the unpolished silicon surface, we introduce an easy approach to fabricate high-performance flexible pressure sensors with irregularly patterned microstructured electrodes. The CNWs microstructures endow the pressure sensor with ultrahigh sensitivity, which is 6.64 kPa-1 in the pressure range <0.2 kPa, 1.26 kPa-1 in the range 0.2 kPa to 1 kPa and 0.15 kPa-1 at 10 kPa. The sensor also presents an ultralow limit of detection of 0.6 Pa due to the surface modification of PDMS by carbon nanowalls. Furthermore, the response time is as short as 30 ms. These indicators show the pressure sensor has an excellent comprehensive performance compared with other pressure sensors with patterned surface. The pressure sensor exhibits immense potential in detection of physiological activities and can be used as human healthcare monitoring. Our tactile sensor arrays can be designed in a variety of shapes to suit different application needs. In future smart health devices, our tactile sensor arrays can be placed in the insole for gait detection and gait pressure distribution image. The characteristics of high-sensitivity, flexibility, and good repeatability provides the possibility to fit on smart wearable devices in the future applications.

Reviewer 2 Report

This manuscript present a highly sensitive, simply fabricated wearable resistive pressure sensor  based on three-dimensional microstructured carbon nanowalls (CNWs)  embedded in a polydimethylsiloxane (PDMS) substrate.

The content and strong point of this manuscript is clear, and the usefulness is also clear. Thus, I think this manuscript is worth to be accepted. However, I think the manuscript has one point that should be corrected.

There are no blank line under the caption of figure, so I think it is a little hard to read manuscript. Thus I hope the point should be considered and the manuscript should be revised.

Author Response

Point 1: The content and strong point of this manuscript is clear, and the usefulness is also clear. Thus, I think this manuscript is worth to be accepted. However, I think the manuscript has one point that should be corrected.

There are no blank line under the caption of figure, so I think it is a little hard to read manuscript. Thus I hope the point should be considered and the manuscript should be revised.

Response 1: Thanks for your kind attention. According to your suggestion, we added a blank line below the caption for each figure and the manuscript was updated carefully.

Round 2

Reviewer 1 Report

All my questions have been addressed. I think paper is much stronger as it is now.